# Experimental and Numerical Investigations into Magnetic Pulse Welding of Aluminum Alloy 6016 to Hardened Steel 22MnB5

**Rico Drehmann** [1,*]**, Christian Scheffler** [2]**, Sven Winter** [2]**, Verena Psyk** [2]**, Verena Kräusel** [2]
**and Thomas Lampke** [1]

1   Materials and Surface Engineering Group, Institute of Materials Science and Engineering (IWW),
    Chemnitz University of Technology, 09125 Chemnitz, Germany; thomas.lampke@mb.tu-chemnitz.de
2   Fraunhofer Institute for Machine Tools and Forming Technology IWU, 09126 Chemnitz, Germany;
    christian.scheffler@iof.fraunhofer.de (C.S.); sven.winter@iwu.fraunhofer.de (S.W.);
    verena.psyk@iwu.fraunhofer.de (V.P.); verena.kraeusel@iwu.fraunhofer.de (V.K.)
*   Correspondence: rico.drehmann@mb.tu-chemnitz.de

**Abstract:** By means of magnetic pulse welding (MPW), high-quality joints can be produced without some of the disadvantages of conventional welding, such as thermal softening, distortion, and other undesired temperature-induced effects. However, the range of materials that have successfully been joined by MPW is mainly limited to comparatively soft materials such as copper or aluminum. This paper presents an extensive experimental study leading to a process window for the successful MPW of aluminum alloy 6016 (AA6016) to hardened 22MnB5 steel sheets. This window is defined by the impact velocity and impact angle of the AA6016 flyer. These parameters, which are significantly dependent on the initial gap between flyer and target, the charging energy of the pulse power generator, and the lateral position of the flyer in relation to the inductor, were determined by a macroscopic coupled multiphysics simulation in LS-DYNA. The welded samples were mechanically characterized by lap shear tests. Furthermore, the bonding zone was analyzed by optical and scanning electron microscopy including energy-dispersive X-ray spectroscopy as well as nanoindentation. It was found that the samples exhibited a wavy interface and a transition zone consisting of Al-rich intermetallic phases. Samples with comparatively thin and therefore crack-free transition zones showed a 45% higher shear tensile strength resulting in failure in the aluminum base material.

**Keywords:** magnetic pulse welding (MPW); AA6016; aluminum; 22MnB5; press-hardening steel; interface characterization

## 1. Introduction

Magnetic pulse welding (MPW), which was initially suggested by Lysenko et al. in 1970 [1], is an innovative technology for manufacturing metallic bonds of similar and dissimilar metals [2]. The setup of the process, which is based on the electromagnetic forming technique [3], consists of the pulsed power generator, the inductor (i.e., the tool), the workpieces to be joined to each other, and additional elements, ensuring that the workpieces are positioned with a defined small gap between them. The pulsed power generator provides storage and a quick release of energy. The most important machine components are the charging units, capacitor banks, high current switches, and control devices [4]. Pereira et al. [5] present an optimized machine, allowing the production of joints with lower energy than commercial machines. The inductor typically consists of a winding, which is embedded in an insulating and reinforcing housing material. Frequently, an additional fieldshaper focuses the acting loads onto the joining zone. Inductor winding and fieldshaper are typically made of a material of high electrical conductivity and acceptable mechanical strength such as CuCrZr or CuBe alloys [6]. Principally, the same machines and very similar tools can be used for MPW, for electromagnetic forming [3], for electromagnetic

impact medium forming [7] or for electromagnetic acceleration of punches, e.g., for the determination of material characteristics up to very high strain rates of $10^4$ s$^{-1}$ [8].

Basically, MPW uses the Lorentz forces of induced currents to accelerate one of the joining partners to velocities at a magnitude of up to several hundreds of meters per second [9]. The impact of this joining partner (the so-called flyer) to the second, typically static, partner (the so-called target) leads to bonding in a defined zone if the collision parameters are appropriate.

Typical process variants of MPW specifically include the welding of tubes to solid [10] or hollow [11] internal parts, as well as the welding of sheets to sheets [12] and sheets to profiles [13]. Welding of tubes to hollow internal parts typically requires a support in order to reduce undesired deformations. This can be a rigid body [11] or an elastomer [14]. New developments in the field of sheet metal welding deal with the adaptation of the technology to spot welding [15].

Figure 1 shows the principle sketch of an exemplary setup for electromagnetic sheet metal welding. Here, an equivalent circuit diagram represents the pulsed power generator. The U-shaped tool coil is connected to this machine so that a damped sinusoidal current flows through the coil when the high-current switch is closed and the capacitor battery is discharged. Due to the different widths and cross section shapes of the branches of the coil, the current density differs significantly. It is high in the narrow branch (i.e., the active side of the inductor), which is positioned close to the area of the workpiece that is to be welded and significantly lower in the wider branch (i.e., the passive side of the inductor). Consequently, the magnetic field, the induced current in the workpiece, and the acting Lorentz forces are much higher in the region of the active side of the inductor when compared to the region of the passive side of the inductor. The Lorentz force ratio corresponds to the inverse ratio of the local conductor width. This strategy for adjusting the force distribution acting on the workpiece was initially suggested in [16] and applied to the electromagnetic forming of sheet metal in [17] and to an electromagnetic tube forming process in [18]. A very similar coil for MPW of sheet metal is used in [19]. The Lorentz forces acting on the workpiece initiate the deformation of the flyer sheet. In the case depicted in Figure 1, the edge of the workpiece bends downwards towards the target and impacts on the target, when it overcomes the initial distance between the two joining partners. Starting from the first contact in the area of the flyer edge, the flyer aligns to the target and the collision point moves along the surface with a collision point velocity $v_c$. During this process, the impact angle $\alpha$ and the impact velocity $v_p$ vary with the position of the collision point [20]. The equations in Figure 1 describe the correlations between the impact angle, the impact velocity, the normal impact velocity $v_n$ and the collision point velocity. If these collision properties correspond to a material-specific process window, a weld is formed, which frequently, but not necessarily, features a wavy character [21].

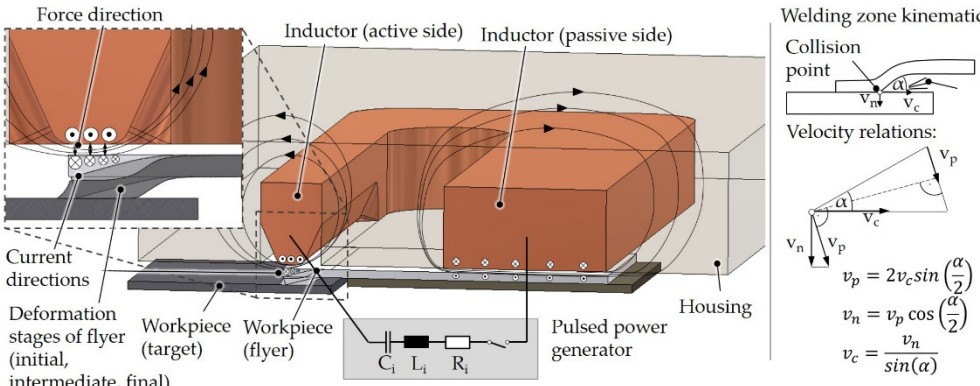

**Figure 1.** Electromagnetic sheet metal welding process setup and process parameter kinematic.

As a solid-state welding process, MPW avoids or at least significantly reduces the typical temperature-induced problems of the more conventional fusion welding processes, such as thermal softening or the formation of intermetallics. Therefore, it is especially promising for joining material combinations usually considered difficult to weld or non-weldable [22]. In his review on electromagnetic pulse welding, Kang presents numerous examples of material combinations that have been successfully joined by MPW [23]. In order to reach high process efficiency and avoid extreme loading of the tool and machine components, the flyer should preferably be made of a material with high electrical conductivity and moderate strength. Therefore, a lot of fundamental research has been dedicated to the MPW of aluminum and copper over recent years. These materials are highly interesting, e.g., for applications in the electrical industry including the currently highly relevant fields of development in battery technology and electro-mobility and in the fields of heating, cooling, air conditioning, and ventilation. Raoelison et al., for example, investigated the interface of aluminum/copper welds and compared it to aluminum/aluminum welds. They found that aluminum/copper welds form intermediate phases in the form of layers or pockets. Depending on the thickness of these phases, the interface becomes sensitive to microcracks and fragmentation [24]. Psyk et al. provide a combined numerical and experimental process analysis for electromagnetic pulse welding of Cu-DHP and EN AW-1050 and consolidate the results in a quantitative collision parameter based process window [20]. They proved that the forces transferable by these joints under lap shear loading can be higher than the maximum forces transferable by the weaker joining partner so that failure occurs outside and far away from the joining region. Wu and Shang have investigated the influence of the surface condition on the weld formation and quality and found that surface scratches in a tangential direction were in favor of a good weld with high strength, while oil on the surface prevented welding [25].

Especially with regard to applications in the automotive industry, another focus of the research on MPW was put on aluminum/steel joints [26]. Kimchi et al. showed that the stand-off distance between an Al tube and a steel bar is a dominant factor for achieving a sound weld and that adding receding angles to the bars can improve weldability [27].

Aizawa et al. provide a detailed study considering MPW of sheets made of multiple aluminum alloys, specifically AA1050, AA2017, AA3004, AA5182, AA5052, AA6016, and AA7075, to cold rolled carbon steel (SPCC) sheets. They considered different machine parameters and coil variants, estimated the corresponding collision velocity and characterized the resulting weld quality via micrographic investigations and lap shear tests [2]. Yu et al. investigated MPW of AA3003-O and steel 20 (0.2 wt.% C) tubes. They also showed that the tension and torsion strength values of the joint are higher than those of the aluminum tube when proper process parameters are chosen [28]. Complementing microstructural investigations indicate that the metallurgical joint composes of two interfaces, a non-uniform transition zone and basic metals with high-density dislocations and nanocrystals. The transition zone features high micro-hardness multi-direction micro-cracks and micro-apertures. Psyk et al. analyzed the influence of adjustable process parameters, specifically the capacitor charging energy, the initial gap width, the lateral position of inductor and flyer and the flyer thickness on collision conditions and the resulting weld quality in terms of transferable force, joint resistance, and weld width for different material combinations, including aluminum/copper and aluminum/stainless steel [29]. Although most of the publications dealing with MPW of steel and aluminum still consider relatively soft steels, recently some attempts have been made to transfer the technology to target materials of high strength. Wang et al. considered MPW of 3003 aluminum alloy sheets and HC340LA steel sheets (zinc-coated and non-galvanized) [30]. They found that although the zinc layer on the galvanized steel was partly removed due to the jet, the remains cause the formation of brittle and hard phases on the interface, resulting in the generation of welding defects, thereby reducing of mechanical properties of the joint. In [31], Psyk et al. exemplarily show that for press hardening steel (22MnB5) and aluminum, also, joining by MPW is basically possible. However, the specific aluminum alloy considered here (EN AW-1050)

has no technological relevance with regard to structural components. In most publications, materials with a low strength and good electrical conductivity such as pure aluminum and copper were joined with MPW, whereas joining high-strength aluminum alloys to hardened steel is extremely challenging and pushes the process to its technological limits. Therefore, it is of great interest that high-strength aluminum alloys can be joined to hardened steel so that novel structural components can be provided for automotive applications.

Therefore, this paper aims for the first time at joining the technologically relevant aluminum alloy AA6016, especially used for structural components, to the press harden-ing steel 22MnB5 (in hardened state) by means of MPW. Sheet metal welding tests are performed and the weld quality is characterized via lap shear tests and microstructural investigations. In parallel, numerical simulations are carried out in order to quantify the conditions of the material collision during welding. Experimental and numerical results are consolidated as a process window applicable for this specific material combination. Deeper understanding of the bonding mechanisms and the microstructural effects in the joining zone is gained by comprehensive microstructural analysis in terms of light micro-scopic investigations, scanning electron microscopy (SEM), and energy-dispersive X-ray spectroscopy (EDX).

## 2. Materials and Methods

Experimental investigations and numerical simulations were carried out consider-ing the aluminum alloy AA6016 T4 and press hardened steel 22MnB5. The chemical compositions of the joining partners as determined by EDX are shown in Table 1.

**Table 1.** Chemical composition (in wt.%) of flyer (AA6016) and target material (22MnB5) according to EDX measurements.

|        | Al   | Fe   | Mn  | Si  | Mg  |
|--------|------|------|-----|-----|-----|
| **AA6016** | 97.6 | 0.4  | 0.1 | 0.6 | 1.3 |
| **22MnB5** | 0.1  | 98.2 | 1.4 | 0.3 | -   |

For industrial processing, 22MnB5 sheets are often protected by an AlSi anti-scaling coating, which can be assumed to have an influence on the weldability of the material combination. However, in order to reduce the complexity, this initial study on MPW of press hardening steel considers uncoated, fully martensitic 22MnB5 sheets. These were austenitized at 950 °C for 8 min in a protective atmosphere of argon and then quenched between cooled steel plates, which avoids distortion of the workpiece due to tempering. In order to ensure a defined homogeneous condition of the AA6016 raw material, which is highly susceptible to aging, the material was solution-annealed, water-quenched, and naturally aged for two weeks. The resulting condition is referred to as T4. The sheets were processed directly after ageing.

In order to describe the process parameters suitable for MPW independently of the specific setup and the properties of the used equipment (pulse power generator, tool inductor), the process window is defined via the local collision parameters, specifically the impact velocity and impact angle $\alpha$ as suggested, e.g., in [32]. In practice, these local process parameters can only be set and changed indirectly via adjustable process parameters such as the capacitor charging energy $E$, the initial gap width between flyer and target $g_{initial}$, or the relative lateral position of flyer edge and center of the active branch of the coil $x_{flyer}$ [20]. Therefore, these parameters were systematically varied in welding tests using the setup shown in Figure 2a. Table 2 provides an overview of the most relevant process parameters characterizing these tests. The experiments were carried out using a pulsed power generator PS103–25 Blue Wave by PST Products (Alzenau, Germany). It features a maximum capacitor charging energy of 103 kJ, a maximum capacitor charging voltage of 25 kV, a stepwisely adjustable capacitance of 25.6 µF to 320 µF, a maximum discharging current of 2.2 MA in the short-circuit, and a maximum short-circuit frequency of 60 kHz. The applied tool coil was self-developed by Fraunhofer IWU. It is intended for producing

a single weld seam directly under the center branch of the coil. This means that only this section is active, while the two outer branches serve for conducting the current back to the connector and the pulsed power generator, so that the current circuit is closed. The used experimental setup is shown in Figure 2c.

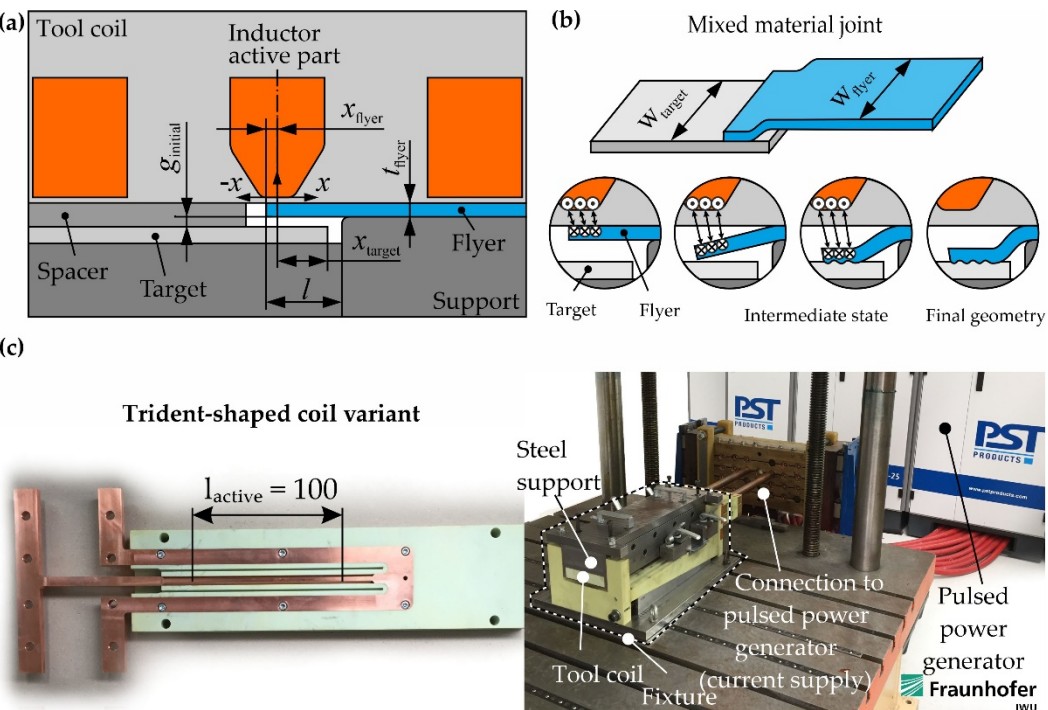

**Figure 2.** (**a**) Scheme of experimental setup; (**b**) Process states during the welding step; (**c**) Real MPW setup with the used trident-shaped coil and the pulsed power generator.

**Table 2.** Parameters of the experimental MPW process.

| Generator Parameters | |
| --- | --- |
| Capacitor charging energy $E$ | 30–40 kJ |
| Capacitance $C$ | 330 µF |
| **Tool parameter** | |
| Active length of inductor $l_{active}$ | 100 mm |
| **Workpiece parameters** | |
| Flyer thickness $t_{flyer}$ | 2 mm |
| Target thickness $t_{target}$ | 2 mm |
| Width of flyer and target | 100 mm |
| **Experimental setup parameters** | |
| Initial gap flyer to target $g_{initial}$ | 0.5–2.5 mm |
| x-position of flyer edge $x_{flyer}$ | −3–0 mm |
| x-position of target edge $x_{target}$ | fixed to 14 mm |
| free length $l$ | fixed to 16 mm |

Three sheets were joined per parameter set. In order to evaluate the weld quality, three lap shear tests were carried out for each joined sheet. In order to avoid failure close to the clamping area during the test, waisted specimens, similar to typical tensile specimens, were prepared from the welded sheet samples as shown in Figure 3a. Welded sheets typically feature non-welded edge zones, because here the induced current turns to form a closed current loop in the workpiece and the magnetic field lines and the Lorentz forces

are re-directed correspondingly, so that the local conditions are inappropriate for welding and can cause local deformations of the flyer edges (see Figure 3a). In order to exclude this effect from the weld evaluation, three specimens per sheet sample were taken at a sufficient distance from the sample edge. The weld quality criterion (evaluation criterion) is based on the failure mode of the lap shear tests specimen, as illustrated exemplarily in Figure 3b. Weld quality is defined as high, if failure occurs in the base material of the weaker joining partner frequently far away from the joining zone (cohesive failure). The weaker partner can be either the one featuring lower material strength or significantly lower wall thickness. In the investigations considered here, it is the aluminum. In contrast, weld quality is defined as critical, if failure occurs in the joining zone by detachment of the two joining partners (adhesive failure).

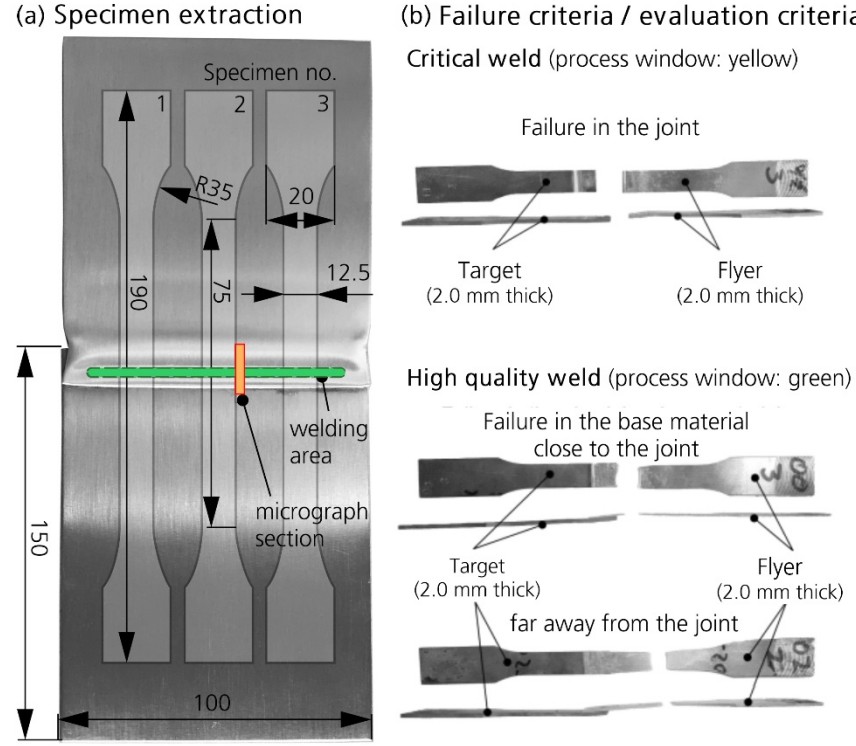

**Figure 3.** (**a**) Specimen extraction from welded samples; (**b**) Different failure modes of the lap shear specimens.

In addition to the global evaluation of the weld quality via lap shear tests, optical microscopic investigations were carried out on a representative section from the welded samples (see Figure 3a). Here, specifically the widths and the precise positions of the welded cross sections were identified, because it is known from literature that the contact zone of a magnetic pulse welded joint typically features welded and non-welded sections [19].

Parallel to the experimental tests, a macroscopic coupled electromagnetic and structural mechanical simulation was used in order to determine the corresponding collision parameters. The modelling was realized in LS-DYNA using the well-established FEM-BEM solver. The electromagnetic model contains the inductor, the flyer, and the target. The material is modelled as linear electromagnetic, and B-H-nonlinearity is disregarded. In the mechanical model, additional necessary supporting elements are considered. Inductor and supporting elements are modelled as rigid bodies, while elastoplastic deformation of flyer and target is possible. Here, an elastoplastic constitutive law with von Mises yield locus, isotropic hardening, and a tabulated scaling factor of the quasi-static flow curve (taken from literature: AA6016 [33] and 22MnB5 [34]), depending on the strain rate, is

applied. The elastoplastic constitutive relation (Equation (1)) of the AA6016 material was assumed in the MPW process simulations as J2-plasticity (von Mises model), considering a strain rate sensitivity $f_c$ and an approach for the material damage $f_c(\eta, \dot{\varphi}, D)$ scaling the quasi-static flow stress $\sigma_0$ in the model of the flow stress $\sigma_y$.

$$\sigma_y(\varphi, \dot{\varphi}, \eta, D) = \sigma_0(\varphi) \cdot f_c(\dot{\varphi}) \cdot f_d(\eta, \dot{\varphi}, D) \tag{1}$$

Here, $\eta$ is the stress triaxiality $\eta = \frac{-p}{\sigma_v}$, calculated by the hydrostatic pressure $p$ and the equivalent stress $\sigma_v$, $\varphi$ the true strain, $\dot{\varphi}$ the strain rate and D the damage parameter (0 . . . 1, 0 = no damage, 1 = complete damage, i.e., element erosion). The damage approach based on the GISSMO model [35] and was used for the MPW simulations simplified as failure model considering an identified relatively high failure strain $\varepsilon_f$ for the existing pressure stress state in the vicinity of the welding zone. Formula (2) describes a logarithmic approach (as known from the Johnson–Cook model) of the strain rate sensitivity, where c is the parameter identified as c = 0.01 for the AA6016 aluminum alloy and $\dot{\varphi}_0$ the reference strain rate (strain rate during quasi static stress strain curve testing, $\dot{\varphi}_0 = 0.001 \text{ s}^{-1}$).

$$f_c = \begin{cases} 1 + c \cdot ln\left(\frac{\dot{\varphi}}{\dot{\varphi}_0}\right) if \ \dot{\varphi} > 10^{-3} \text{ s}^{-1} \\ 1.0 \ if \ \dot{\varphi} \leq 10^{-3} \text{ s}^{-1} \end{cases} \tag{2}$$

The discretization size has been optimized considering the contradicting requirements related to accuracy of the calculation result and calculation time. The result is shown in Figure 4. In order to allow a realistic modelling of the current distribution and the penetration depth of the EM field, especially the elements of the tool coil that are close to the coil surfaces facing the workpieces must feature small size and it is necessary to model several elements over the thickness of flyer and target. Here, six elements with an initial element height of 0.5 mm were used. With regard to the mechanical calculation model, fine discretization is needed in those areas of flyer and target that are deformed during the welding process. Therefore, in the workpiece areas that are close to the active branch of the coil, also the initial element width was set to 0.5 mm, while it is higher in the undeformed areas far away from the active branch of the coil.

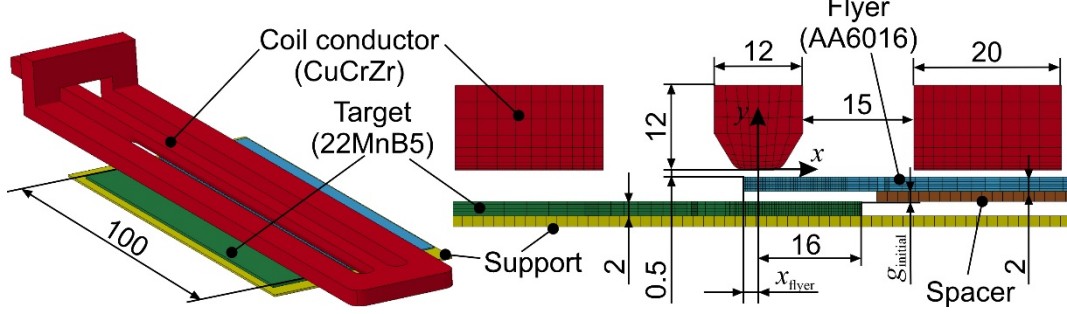

**Figure 4.** Discretization of the numerical model.

Obviously, the overall result of a coupled simulation significantly depends on the update time step, i.e., the duration between two subsequent recalculations of the electromagnetic system. Here, a time step of 50 ns was used. The aim of the simulation was to identify the collision parameters (impact velocity and impact angle) during the MPW. For this, however, it was not necessary to consider the influence of temperature during the simulation.

Finally, the experimental and the numerical results were consolidated as a collision parameter-based process window. For this purpose, local collision parameters are determined numerically and the position of welded and non-welded specimen sections determined via light-microscopic investigations are correlated to each other. A process

window in the form of a diagram showing the impact velocity on the abscissa and the impact angle on the ordinate is presented. Collision parameters of welded specimen sections are indicated as welded (i.e., inside the process window), while collision parameters of non-welded sections and of completely non-welded specimens are indicated as not welded (i.e., outside the process window). Collision parameters of welded specimen sections are further distinguished into critically welded and high-quality welded ones according to the result of the corresponding lap shear test, specifically the failure mode. Thus, the process window was constructed by combining the results from the different process parameters, the numerical simulations, the microstructural investigation of the weld zone, and the lap-shear test.

To get a deeper understanding of the microstructural effects occurring during electromagnetic pulse welding, microstructural investigations of the aluminum/steel joints included scanning electron microscopy (SEM) in secondary electron (SE) and backscatter electron (BSE) mode that was accompanied by energy-dispersive X-ray spectroscopy (EDX). EDX investigations comprised point analyses as well as EDX mappings of the joining zone, executed with a field emission SEM Zeiss NEON 40EsB (Zeiss AG, Oberkochen, Germany). The respective cross sections were prepared with a final oxide polish (OP-S) using vibrational polishing (Buehler Vibromet 2, Mastertex, low nap, 60 min).

Furthermore, nanoindentation measurements according to ISO 14577 were conducted in order to determine the hardness of the area that is close to the interface of the aluminum/steel joint. The measurements were carried out on a UNAT nanoindentation device by ASMEC GmbH/Zwick GmbH & Co. KG (Dresden/Ulm, Germany) with a Berkovich indenter (0.394 μm tip radius). The normal force was increased from 0 to 5 mN within 10 s, followed by a hold time of 5 s and a force relief period of 4 s.

## 3. Results and Discussion

### 3.1. Determination of MPW Process Window

The result of the numerical process modelling in terms of quantified collision parameters is exemplarily shown in Figure 5. Here, a high-quality weld is achieved with a capacitor charging energy $E$ of 40 kJ, an initial gap width $g_{initial}$ of 2.5 mm, and a lateral relative position of the coil center and the flyer edge $x_{flyer}$ of 0 mm. All other process parameters correspond to the fixed process parameters given in Table 2. In the following, this parameter set is referred to as condition 1. Both velocity curve and angle curve are depicted as functions of the distance $d$ from the flyer edge in Figure 5. They show a typical shape that is representative of all investigated parameter combinations. The velocity curve features a plateau of relatively constant velocity, which is followed by a steady decline of the curve. In this specific case, the plateau is rather short (approx. 2 mm) and high velocity (approx. 600 m/s) is reached because of the high capacitor charging energy in combination with a high initial gap width, which corresponds to a long acceleration distance for the flyer. With other parameter combinations considered here, the plateau reached lengths of up to 6 mm at a lower height. The subsequent steady decline of the curve is about equally steep for all regarded parameter combinations.

In contrast, an initial rising trend, which is then reversed to a decrease characterizes the angle curve. The high initial gap in combination with the short lateral relative position of the coil center and the flyer edge leads to a relatively steep rise of the curve and high absolute values as compared to other parameter combinations considered in this study. These observations are in good agreement with the general trends and influences related to the collision conditions presented in [29].

Additionally, Figure 5 directly correlates the velocity and angle curves with the result of the investigations by optical microscopy. The welded area identified in the microstructure was projected onto the diagram so that velocity-angle combinations that are beneficial for weld formation can easily be identified. In the next step, this information was transformed to collision parameter-based process windows. Figure 6 consolidated the results of all welding experiments, the corresponding numerical simulations and the evaluation of the

weld quality. Collision parameter combinations corresponding to the welded zone were regarded as high-quality weld (green points) or critical weld (yellow points) depending on their failure mode (see Figure 3) and the maximum reached forces in the lap shear test (Table 3), whereas collision parameter combinations corresponding to the non-welded zone are depicted as red crosses. Green points can be considered safely within the process window, yellow points indicate the edge of the process window, and red crosses can be considered outside the process window.

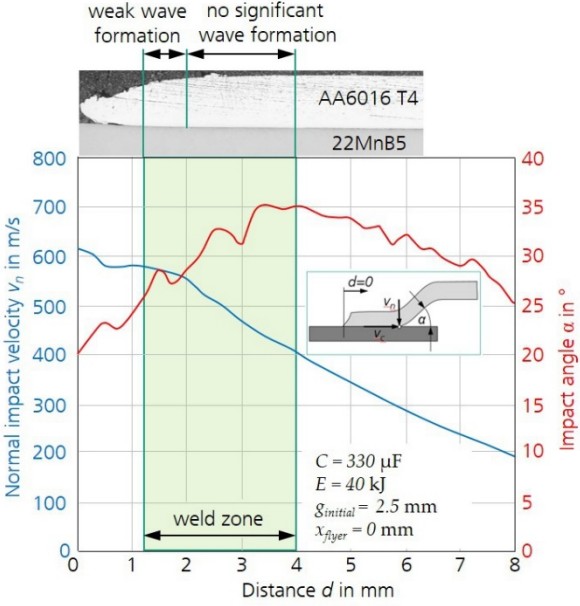

**Figure 5.** Comparison of the collision parameter distribution for a high-quality welding sample (condition 1).

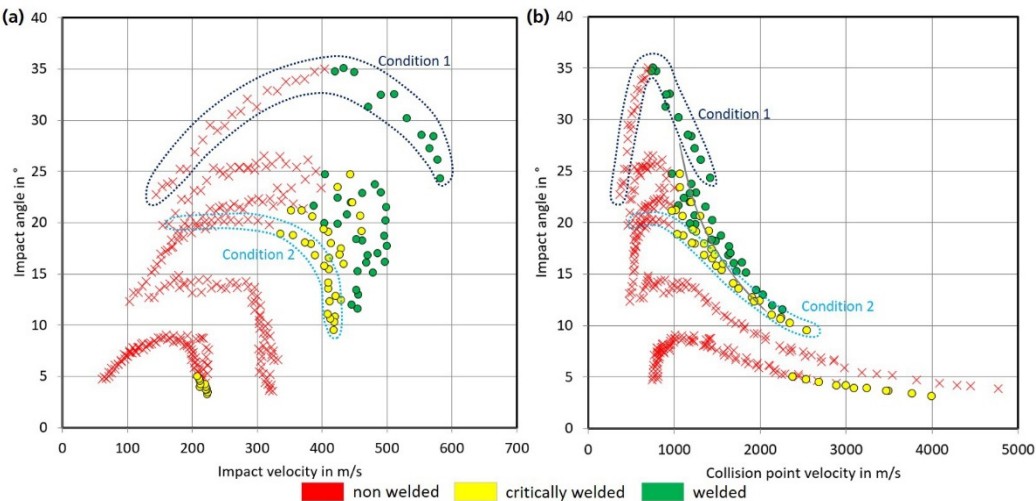

**Figure 6.** Identified process window for MPW of AA6016 T4 to 22MnB5 (hardened) (**a**) based on normal impact velocity $v_n$ and impact angle $\alpha$, (**b**) based on collision point velocity $v_c$ and impact angle $\alpha$. The dotted lines indicate the collision parameters of the high-quality (condition 1; dark blue line) and the critical weld (condition 2; light blue line).

**Table 3.** Results of the lap shear test and resulting failure mode for Condition 1 and Condition 2.

|  | Maximum Force *F* in N | Displacement at *F* in mm | Failure Mode |
|---|---|---|---|
| **Condition 1** $E = 40 \text{ kJ}$ $g_{initial} = 2.5 \text{ mm}$ $x_{flyer} = 0 \text{ mm}$ | $2813 \pm 73$ | $5.11 \pm 0.31$ | **high-quality weld** (fail in base material) |
| **Condition 2** $E = 35 \text{ kJ}$ $g_{initial} = 1.5 \text{ mm}$ $x_{flyer} = -2 \text{ mm}$ | $1898 \pm 56$ | $1.86 \pm 0.18$ | **critical weld** (fail in joint) |

As the collision parameters were determined at discrete points with a distance of 0.25 mm between them (corresponding to the element size in this section of the numerical model), welding tests with one specific parameter set deliver a number of points in the process window. Figure 6 exemplarily highlights the group of points delivered by the welding samples processed with the parameter set referred to as condition 1. Additionally, a second group of points delivered by welding tests performed with a capacitor charging energy of 35 kJ, an initial gap of 1.5 mm between flyer and target, and a lateral relative position of the coil center and the flyer edge of −2 mm is indicated as condition 2. This parameter set was found to lead to critical welds failing by detachment of flyer and target in the joining zone und lap shear load.

In this context, the interpretation of the experimental results in the area close to the flyer edge is known to be challenging. Here, attachments of the flyer and target material to the respective other joining partner as exemplarily shown in [20] might indicate that the area was initially welded and ripped open again during the MPW process. However, as this is not fully clear, a zone of up to 1.3 mm distance from the flyer edge was disregarded when composing the process windows in order to avoid incorrect assignment either to collision parameters leading to a welded or non-welded contact.

Figure 6 shows two different variants of collision parameter-based process windows. They consider the impact angle $\alpha$ as a function of the normal impact velocity $v_n$ (Figure 6a) and the collision point velocity $v_c$ (Figure 6b), respectively. The equations for these velocities are explained in Figure 1. The latter is more common, especially in the context of explosive welding, an alternative impact welding technology, which shows some similarities to MPW although it is based on a chemical energy source instead of an electric one and the temperature regime during the process differs significantly. Both variants of the process window allow clear identification of collision parameter combinations that can be expected to lead to robust high-quality welds, i.e., failure in the base material of the weaker joining partner and those that will clearly not lead to welding. In-between these two groups there is a diffuse border area. This border area appears more distinct in the process window based on the collision point velocity, but it must be considered that the velocity scale in this diagram is roughly ten times as high compared to the scale in the diagram that is based on the impact velocity. In summary, an impact angle >15° and an impact velocity >400 m/s lead to sufficiently good joints between the aluminum alloy AA6016 and the steel 22MnB5.

### 3.2. Detailed Microstructural Characterization

In addition to the goal of determining a process window for MPW of AA6016 to 22MnB5, another objective of this work was the detailed investigation of the joining zone in order to gain deep knowledge about the microstructural effects in the joint zone. Specifically, differences between a high-quality weld and a critical weld were to be identified. For this purpose, the weld samples referred to as condition 1 (high weld quality) and condition 2 (critical weld quality) were regarded further. In this respect, already the examinations by optical microscopy revealed a significant difference between these two sample types. By preparing several cross-sections of the Al/steel compound at varying distances from the

edge of the sheets (compare Figure 7a), the approximate width and length of the joining zone, i.e., the zone of intimate material contact between Al and steel, were determined. Figure 7b,c exemplarily show position 4 for the high-quality weld (weld parameters according to condition 1) and the weld of critical quality (weld parameters according to condition 2), respectively.

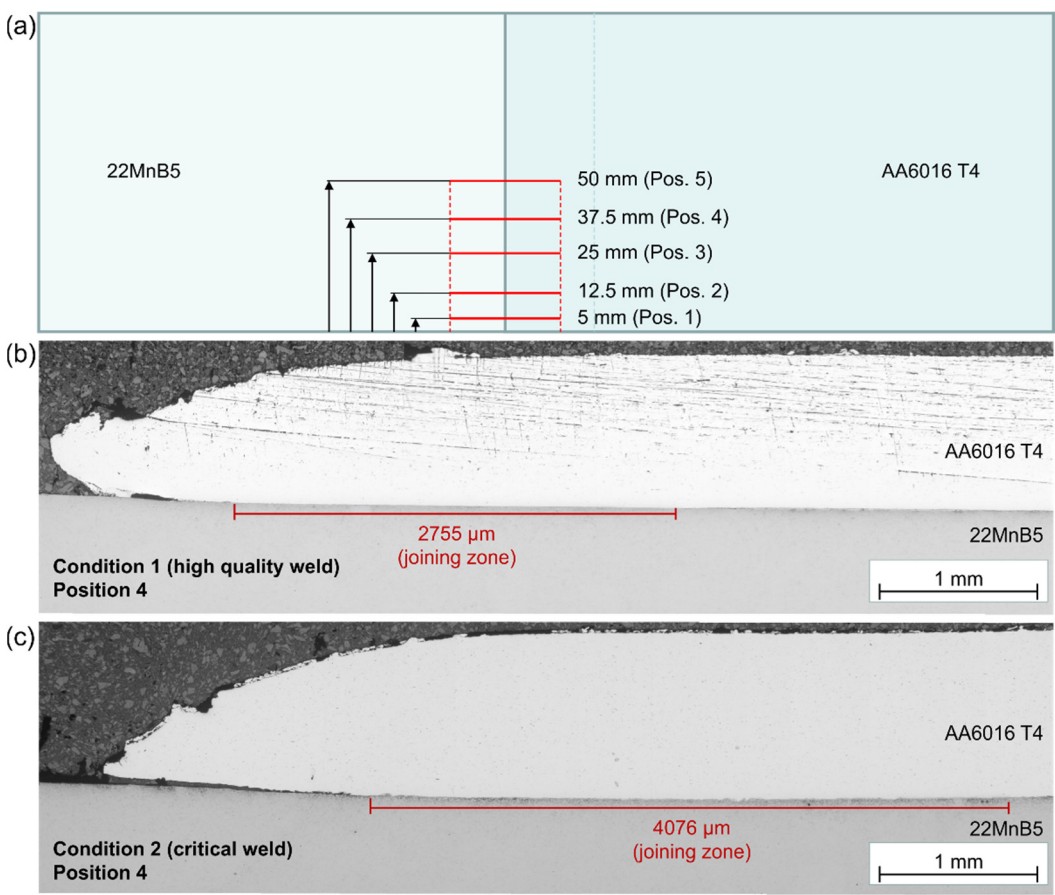

**Figure 7.** (**a**) Positions of cross-sections at varying distances from the edge of the Al/steel compound. (**b**) Optical microscope image of the cross-section of a high-quality weld (condition 1, position 4), (**c**) Optical microscope image of the cross-section of a critical weld (condition 2, position 4).

As confirmed by several high-quality and critical welds and to the surprise of the authors, the area of the joining zone of a high-quality weld is roughly half as large as that of a critical weld. In order to visualize this, the measured joining zone widths at positions 1 to 5 were schematically transferred to a macroscopic top view image of the Al/steel compound. The resulting assumed shape of the joining zone of both a high-quality and a critical weld is shown in Figure 8, illustrating the much smaller joining zone of the high-quality weld.

Since this observation seemed to contradict the lap shear test results and correlations observed in earlier studies on different material combinations [29], the joining zone was thoroughly investigated by SEM. This characterization revealed features that are typical for MPW joints, namely the formation of a wavy interface in the contact area of the metal sheets and the existence of an interface-near transition zone supposedly consisting of intermetallic phases. As shown in Figure 9a, the intermetallic phases (IMP) are predominantly present in the valleys of the wavy interface. Their existence is considered evidence for the temporary melting of flyer and target material in a very narrow interfacial zone due to the impact-induced conversion of the kinetic energy of the flyer into thermal energy.

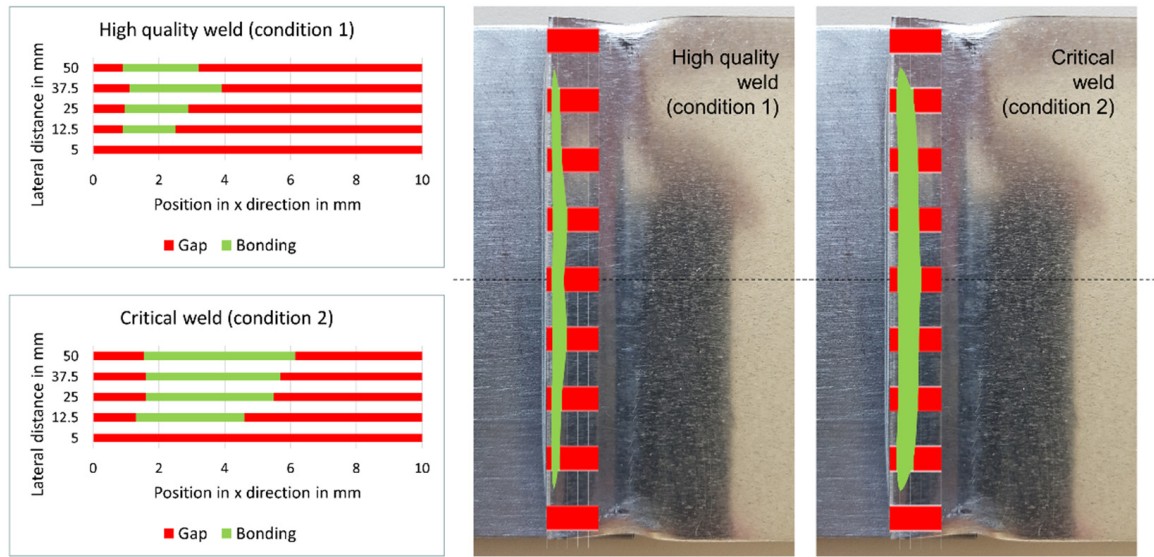

**Figure 8.** Measured width and assumed shape of the joining zones of a high-quality weld and a critical weld in order to visualize the significantly larger joining area (marked green) of the critical weld.

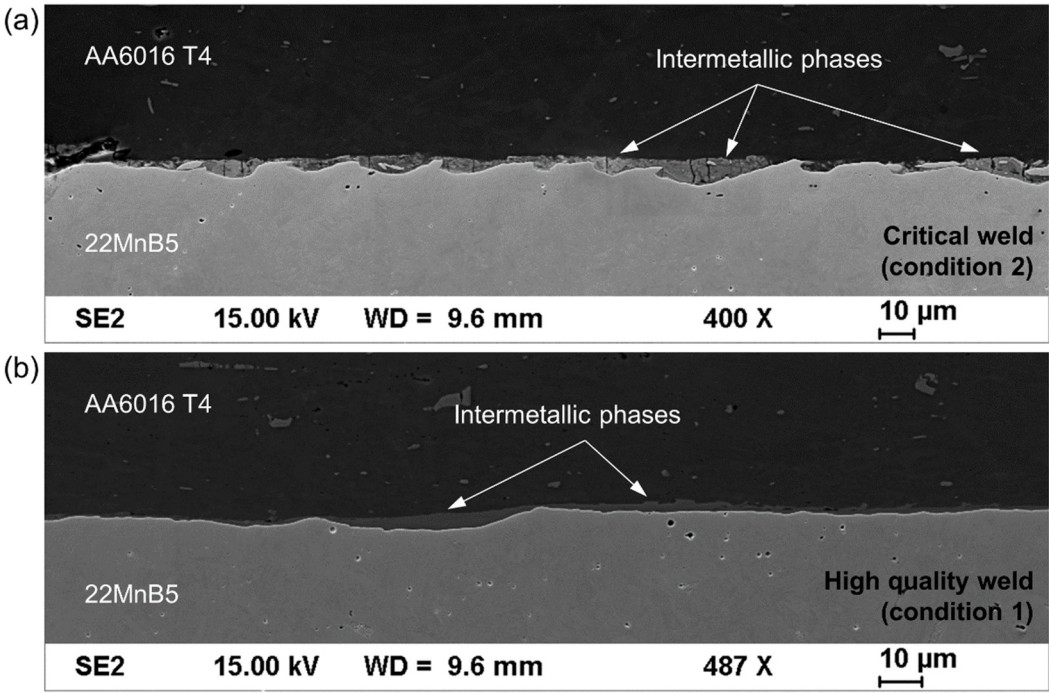

**Figure 9.** (**a**) SEM image (SE mode, acceleration voltage 15 kV, working distance 9.6 mm) of the joining zone of AA6016 with a fully martensitic 22MnB5 steel (critical weld). (**b**) SEM image (SE mode) of the joining zone of AA6016 with 22MnB5 steel (high-quality weld).

The comparison of the cross-section of a critical weld (parameter set during welding according to condition 2) in Figure 9a with a high-quality weld (parameter set during welding according to condition 1) shown in Figure 9b reveals a significantly less pronounced wave formation for the latter. Areas with IMP are still visible, although the transition zone is much thinner and IMP occur less frequently over the entire cross-section.

A possible explanation for the interface failure of the critical welds in the lap shear test despite the larger joining zone is provided by higher magnification SEM images. In Figure 10, the transition zone between Al and steel exhibits vortex-like structures, indicating a strong mixing of flyer and target material during the joining process. However, the formed IMP area shows numerous, mostly vertical cracks that extend over the entire transition zone and end abruptly in the adjacent Al and steel material. Since these cracks within the obviously very brittle IMP occur over the entire joining zone, this pre-damage is very likely the reason for the interfacial adhesive failure of the critical welds in the shear tensile test. In contrast, the high-quality welds, which exhibit almost no cracks in the transition zone, showed a cohesive failure in the Al sheet, thus indicating a superior interface bonding strength in comparison to the critical welds even though the joining zone is considerably smaller.

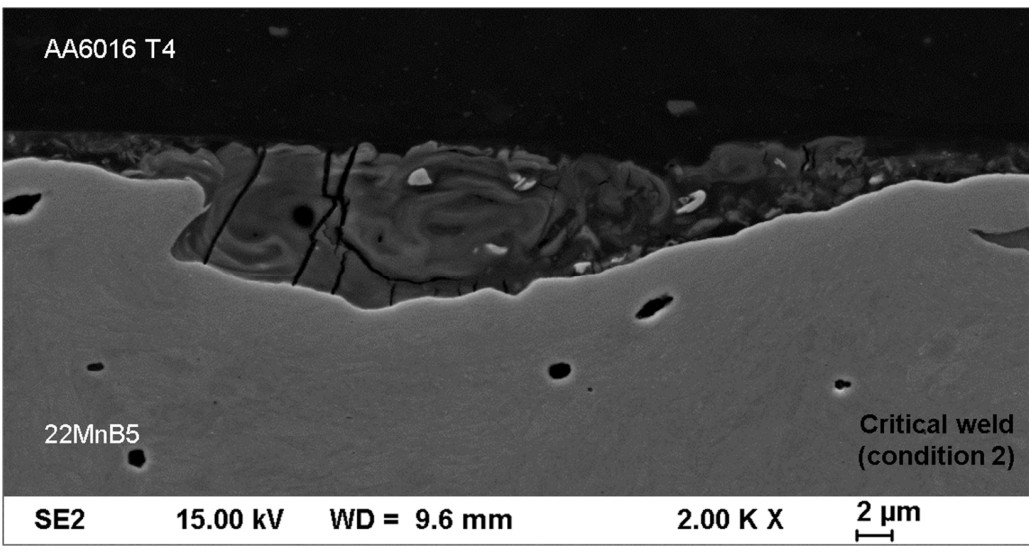

**Figure 10.** SEM image (SE mode) of the transition zone of a critical weld (welding parameters according to condition 2) with numerous cracks.

Further investigations were conducted in order to identify the IMP in the transition zone. An EDX mapping of an exemplarily chosen part of the interfacial area of a critical weld (Figure 11) revealed that the transition zone mainly contains Al, although the Al content seems to vary in a certain range as confirmed by the heterogeneous distribution of Fe, Mg, Mn, and Si in this zone. Additionally, oxygen can be detected, but is predominantly found in the described cracks in the transition zone.

In the SEM image shown in Figure 12a, which was taken in backscattered electrons mode, many different grayscales in the transition zone of a critical weld can be observed, indicating an inhomogeneous chemical composition. In contrast, the gray shade in the transition zone of the high-quality weld is much more homogeneous (see Figure 12b), thus suggesting a correspondingly homogenous chemical composition.

EDX point analyses at different locations in the transition zone (spots 1 to 3 in Figure 12a and 4 to 5 in Figure 12b) verify this observation. Table 4 summarizes the corresponding results. They confirm that the transition zone of the critical weld consists of Al-rich intermetallic phases with a strongly varying Al/Fe ratio and Mg, Si and Mn as accompanying elements ($\leq 1$ at.% each). A comparison with the known Al-rich iron aluminide phases shows that $FeAl_2$ (66.7 at.% Al), $Fe_2Al_5$ (71.4 at.% Al), and $Fe_4Al_{13}$ (76.5 at.% Al) might have been formed in the Al/steel interface during the joining process. However, due to the heterogeneous microstructure, even on a very small scale, a clear assignment of single areas to the mentioned phases is not possible by means of the characterization methods that were used in this work. Analytical methods with a very high spatial resolution such as X-ray photoelectron spectroscopy (XPS) might give more insight into the

phase composition of the transition zone. Taking into account the very short time period of the joining process and the high cooling rates in the joining zone, the transition zone is expected to consist either of nanoscaled metastable phases and/or even amorphous areas.

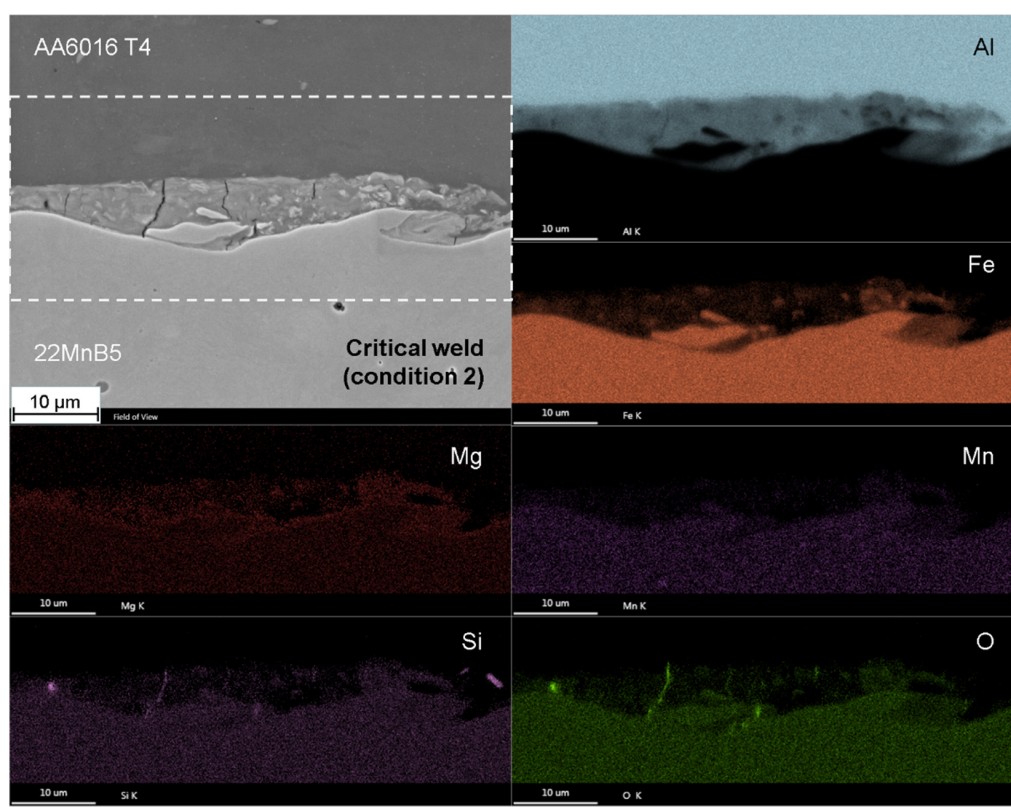

**Figure 11.** SEM image and EDX mapping of the Al/steel interface area of a critical weld (welding parameters according to condition 2), including the elemental distribution of Al, Fe, Mg, Mn, Si, and O.

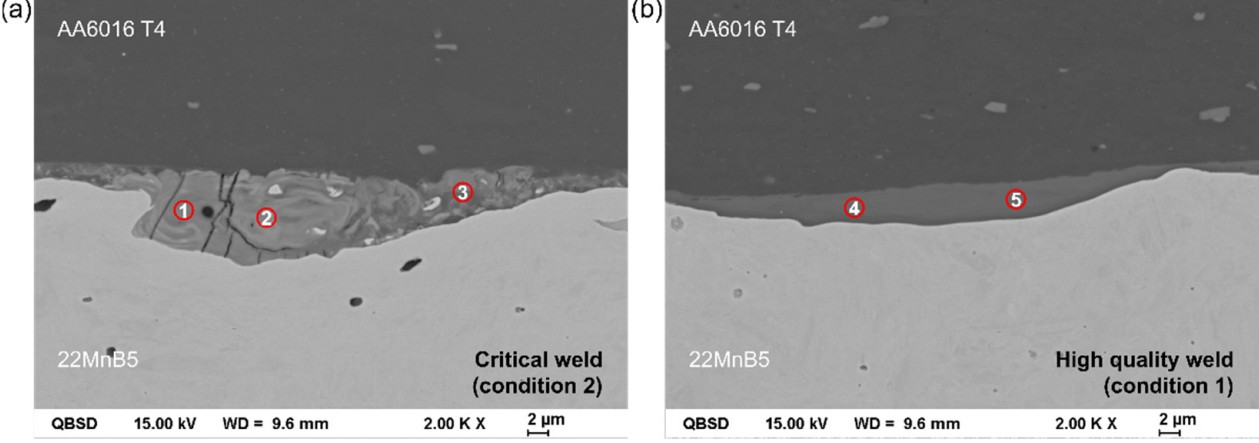

**Figure 12.** SEM images (BSE mode) of the joining zone of (**a**) a critical weld and (**b**) a high-quality weld with marked spots (1 to 5) for EDX point analysis.

In contrast to the critical weld, the transition zone of the high-quality weld (Figure 12b) exhibits a nearly constant chemical composition of about 88 at.% Al and 10 at.% Fe. This was also confirmed by EDX mapping (not shown here), which indicated a very homogeneous Al-rich phase in the transition zone. Regarding the mentioned composition, no stable FeAl phase with such a high Al content is known from the literature, which suggests the presence of non-equilibrium phases or a supersaturated solid solution in the transition zone of the high-quality welds, but partly also in the critical welds (see spot 3 in Figure 12a).

**Table 4.** Chemical composition of spots 1 to 5 from Figure 12 determined by EDX point analysis.

| Element | | Al | | Fe | | Mg | | Si | | Mn | |
|---|---|---|---|---|---|---|---|---|---|---|---|
| | | wt.% | at.% | wt.% | at.% | wt.% | at.% | wt.% | at.% | wt.% | at.% |
| Critical weld | Spot 1 | 60.8 | **75.5** | 37.3 | **22.4** | 0.7 | **1.0** | 0.7 | **0.8** | 0.5 | **0.3** |
| | Spot 2 | 52.5 | **69.0** | 45.6 | **29.0** | 0.6 | **0.8** | 0.6 | **0.8** | 0.7 | **0.4** |
| | Spot 3 | 81.5 | **89.2** | 16.7 | **8.8** | 0.8 | **1.0** | 0.7 | **0.8** | 0.3 | **0.2** |
| High-quality weld | Spot 4 | 79.6 | **87.9** | 18.2 | **9.7** | 1.1 | **1.4** | 0.7 | **0.8** | 0.4 | **0.2** |
| | Spot 5 | 79.6 | **88.0** | 18.3 | **9.8** | 1.0 | **1.3** | 0.7 | **0.7** | 0.4 | **0.2** |

Figure 13 presents the results of the nanoindentation of a high-quality weld. The six hardness values on the Al side (top row) are quite similar and well within the microhardness range for severely plastically deformed AA6016 T4 described in the literature [36], except for the last one, where the indenter obviously hit a precipitation in the Al alloy. Additionally, the six hardness values on the 22MnB5 side (bottom row) are within a relatively narrow range around 480 HV, once again in good accordance with 22MnB5 hardness values known from the literature [37,38]. In contrast, the hardness values in the transition zone (middle row) exhibit a relatively large scatter. As described above, this transition zone supposedly consists of intermetallic non-equilibrium FeAl phases. Investigations by several authors have shown that especially Al-rich FeAl phases, as they were observed here, exhibit very high hardness values (>700 HV) which is usually associated with an increased brittleness [39,40]. Obviously the indents that are farther away from the interface of the transition zone with the 22MnB5 sheet, i.e., the first, fourth and fifth indent (from the left), achieve the lowest hardness values. The closer the indents get to the 22MnB5 side, the higher are the hardness values, reaching a maximum with the second indent (781 HV), which is almost exactly on the interface between the transition zone and the 22MnB5 material. As mentioned before, only high-resolution characterization methods could answer the question if this observed increase in hardness has to be attributed to a change in the phase composition within the interfacial area.

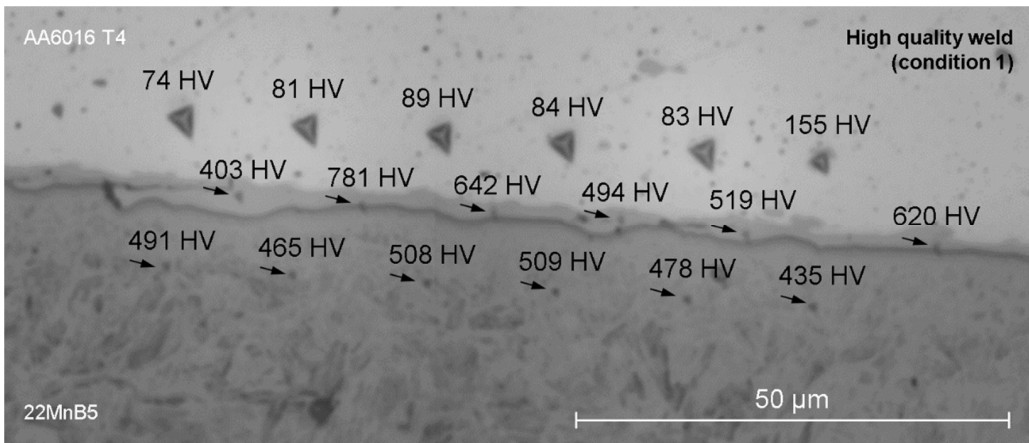

**Figure 13.** Nanoindentation results of the interfacial area of an Al/steel joint (high-quality weld, indentation force 5 mN).

## 4. Summary and Conclusions

AA6016 sheets were successfully joined for the first time to hardened 22MnB5 steel by magnetic pulse welding (MPW). By combining an extensive experimental study with a macroscopic coupled multiphysics simulation in LS-DYNA, a process window for high-quality welds was determined. Subsequent lap shear tests showed either cohesive failure in the AA6016 base material or interface failure in the joining zone. Samples that showed the latter failure mode were classified as critical welds, while those ones failing cohesively in the aluminum base material were evaluated as high-quality welds. For the chosen experimental setup, high-quality welds were formed for normal impact velocities of the AA6016 flyer starting at about 400 m/s at impact angles of around 10°. A tendency was observed that for larger impact angles, higher normal impact velocities are necessary in order to produce high-quality welds.

Microstructural characterization surprisingly revealed significantly larger joining zone areas for critical welds than for high-quality welds. However, next to a pronounced wave formation at the Al/steel interface and vortex-like structures indicating a strong mixing of AA6016 and 22MnB5, the intermetallic transition zone of the critical welds exhibited a lot of vertical cracks, thus providing a possible explanation for the inferior bonding strength in the lap shear tests. By contrast, the high-quality welds were characterized by thinner, but very homogeneous and crack-free transition zones consisting of an Al-rich intermetallic phase with about 88 at.% Al and 10 at.% Fe. Therefore, it can be concluded that for very high normal impact velocities above 400 m/s at impact angles between 10° and 35°, there is a continuous change in the way the transition zone forms. Even though the underlying mechanisms are not yet fully understood and require further research, it can be stated that process parameters that lead to a merely slightly pronounced wave formation and the development of a homogeneous transition zone with a very high Al content appear most suitable for the production of high-quality welds between AA6016 and 22MnB5. In summary, the following key results were determined and are essential for a perspective transfer of hybrid MPW joints made of high-strength aluminum alloys and hardened steel for use as structural components in automotive engineering:

- For the first time, the aluminum alloy AA6016 was successfully joined to hardened steel 22MnB5 by MPW.
- A robust process window for high-quality welds was determined by a macroscopic coupled multiphysics simulation in LS-DYNA.
- Surprisingly, the high-quality welds were characterized by thinner, but very homogeneous and crack-free transition zones consisting of an Al-rich intermetallic phase.

**Author Contributions:** Conceptualization, R.D., C.S., S.W. and V.P.; methodology, R.D., C.S. and V.P.; validation, R.D., C.S. and V.P.; formal analysis, R.D. and C.S.; investigation, R.D. and C.S.; resources, V.K. and T.L.; writing—original draft preparation, R.D., C.S. and S.W.; writing—review and editing, V.P. and T.L.; visualization, R.D. and C.S.; supervision, V.P. and T.L.; project administration, R.D., C.S. and V.P. All authors have read and agreed to the published version of the manuscript.

**Funding:** This research received no external funding.

**Data Availability Statement:** Not applicable.

**Acknowledgments:** The authors would like to express their gratitude to Elke Benedix, Steffen Clauß, Paul Seidel and Christian Loos for their support in sample preparation and microstructural characterization.

**Conflicts of Interest:** The authors declare no conflict of interest. The funders had no role in the design of the study; in the collection, analyses, or interpretation of data; in the writing of the manuscript, or in the decision to publish the results.

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
