# Peer review of "Experimental and Numerical Investigations into Magnetic Pulse Welding of Aluminum Alloy 6016 to Hardened Steel 22MnB5"

_jmmp, doi:10.3390/jmmp5030066_

Round 1
Reviewer 1 Report
The scientific paper proposes experimental and based numerical studies concerning the application of a magnetic pulse welding process. The scientific article is very well structured and contains rigorous and pertinent comments, illustrations and analysis. A very rich literature references are used and inserted along the evolution of presented welded process details, materials, methodologies and qualitative/quantitative results.
A lot of minor revisions are compulsory:
In a first time the paper title must be changed on “Experimental and numerical magnetic pulse welding of aluminum alloy 6016 to hardened steel 22MnB5”. The word “windows” must be replaced along all the text with corresponding context synonym. The authors must mention is the used multi-physics LS-DYNA model are based on non-symmetric Cauchy stress mechanics. In fact The Magnetic momentum generated local momentum densities and the Cauchy-stress in never symmetric. According to the proper and useful diagram on Figure 5 the authors must added during the article part which contain the comments of desired optimal values of impact velocities and orientations (angle “alpha”) the correspondence with précised values of capacitor charging energy E and Capacitance C (well illustrated on Table 2 as input generator variables of welding process). If possible also to add the majority of material parameters, form of constitutive flow laws, values of thermal, elasto-plastic-thermal expansion and mechanical parameters together with the all corresponding boundaries conditions especially concerning the thermal computation.
It is also required to take into account the following corrections:
1° Page 6 - Figure 3 – it is required to explain the white/black observed colour (legend and signification).
2° Page 9 - Figure 6 – it is required to mentioned in the legend Figure what is Condition 1 and what is the Condition 2.
3° Page 14 – Concerning Figure 13 the authors must detailed quantitative comments on the observed orthogonal hardness gradient with a magnitude change of 5 and 7 from the observed first lower values on the outside welding area. Correlation between the thickness values of the presented three different layers (with important orthogonal hardness) with the global welded thickness is also necessary to understand their influence level. In fact the hardness results have important impact on the strength and failure resistance of the materials welding area.
Regarding all above required minor corrections together with the shown recommendations, if all these ones are taken into the account, the paper can be accepted for publication in MDPI Journal of Manufacturing and Materials Processing.
Reviewer 2 Report
The paper entitled “Magnetic pulse welding of aluminum alloy 6016 to hardened 2 steel 22MnB5” presents an interesting study on magnetic Magnetic Pulse Welding of two different materials. For this paper, some considerations should be taken into account:
- Abstract:
Line 23: “showed a superior shear tensile”
Q - How much higher? To specify
- Keywords:
Q - There are many keywords, focus on the main ones.
- Introduction:
Line 69: “the acting Lorentz forces are much higher in the region”
Q - How much higher? To specify
Line 107: “of a good weld”
Q - What do the authors mean by “good weld”? To explain.
Q - The actors present a literature review with a description of the MPW process and some studies of different materials welded by MPW, however, they should make it clearer what the knowledge gap is, that is, what was the work's differential. Was it using a different aluminum?
Line 140: “technologically relevant aluminum”
Q - Relevant for what?
- Materials and Methods:
Q - Authors could add a photograph of the welding device with the parts prepared for welding and/or experimental apparatus.
Q - Numerical simulation is briefly described. Authors could add all the information regarding the simulation (which are the properties of the materials used in the simulation, what are the initial conditions, the equations involved, what mesh is used, etc.).
Q - An experimental design could be presented (how many specimens were produced, which factors, what levels of each factor, etc.). Also, a statistical methodology could be used to corroborate the results.
- Results and Discussion
Q - As the authors guarantee that the numerical simulation corroborates the real (experimental) values. Could the authors compare the results of the numerical simulation on the collision parameters with some experimental result?
Q - Figure 6 should be explored further. What is the difference between the speeds of graphics (a) and (b)? Is there a correlation between these quantities? How could the process be optimized (optimal operating range)?
Q - Mechanical tests for the characterization of the welding were not found in the results and discussions. Review!
Reviewer 3 Report
The reviewer comments of the paper «Magnetic pulse welding of aluminum alloy 6016 to hardened steel 22MnB5»- Reviewer
The authors presented an article «Magnetic pulse welding of aluminum alloy 6016 to hardened steel 22MnB5». However, there are several points in the article that require further explanation.
Comment 1:
Title needs to be rewritten more specifically and clearly. Now from the title it is not clear why it is the article and it investigates? Title should reflect the purpose of the article.
Comment 2:
Abstract.
What are the quantitative and qualitative results obtained? What a scientific novelty and practical significance?
Comment 3:
The introduction of the article should be significantly improved.
Authors need to add a paragraph analyzing the relevance of materials. What is this material? What MPW problems are there?
What are the “white” spots? Why is your research important to science and the reader?
Add article in the introduction:
DOI: 10.1007/978-981-15-5151-2_2
You must prove that scientists did not solve the problem of these studies before.
At the end of the introduction formulate a clear and understandable purpose of the article at the end of the introduction.
Comment 4:
Materials and Methods
For devices and machine used in research, indicate in parentheses (manufacturer, city, country).
How many repetitions are used in the measurement? What statistical methods are used to process experimental results? Please describe this in more detail.
Add the physical and mechanical properties of materials for which the FEM calculation was performed. What are the boundary conditions? Show all this on a design diagram. What PC is used for calculations? What software and why? What is the calculation time? What type of items and why? What assumptions are made? What temperatures are reached? Does this have a negative effect on the weld?
Comment 5:
Results and Discussion
Are all the figures in the article original? If not needed appropriate citations and publisher permissions.
The quality and resolution of all figures needs to be improved.
I would like to see the results of calculations using the FEM. Stress fields, temperature fields, etc. And the corresponding description of these results in the text of the article.
But most importantly, what loads and stresses can the resulting weld be able to withstand? That is, on what basis do the authors conclude that a high-quality weld has been obtained?
Comment 6:
It will be useful to add a section of Nomenclature in which to sign all the physical quantities and abbreviations encountered in the article. There are many physical quantities in the text and such a section will help to find the description of the necessary element.
For example,
MPW : Magnetic pulse welding
etc.
Comment 7:
The conclusions need to be improved.
What is the novelty of the article? What is the practical significance? What are the differences from previous works?
Conclusions should reflect the purpose of the article.
The article is interesting. However, the article needs to be improved. Authors should carefully study the comments and make improvements to the article step by step. All changes should be highlighted in color. After major changes can an article be considered for publication in the "Journal of Manufacturing and Materials Processing".
Round 2
Reviewer 3 Report
The authors have improved the article according to the comments. The article can be accepted for publication.